# Implementation and prospective real-time evaluation of a generalized system for in-clinic deployment and validation of machine learning models in radiology

**James R. Hawkins**[1]*, **Marram P. Olson**[1], **Ahmed Harouni**[2], **Ming Melvin Qin**[2], **Christopher P. Hess**[1], **Sharmila Majumdar**[1], **Jason C. Crane**[1]

**1** Center for Intelligent Imaging, Department of Radiology and Biomedical Imaging, University of California, San Francisco, California, United States of America, **2** NVIDIA, Santa Clara, California, United States of America

* James.Hawkins@ucsf.edu

**Data Availability Statement:** All relevant data are within the manuscript.

## Abstract

The medical imaging community has embraced Machine Learning (ML) as evidenced by the rapid increase in the number of ML models being developed, but validating and deploying these models in the clinic remains a challenge. The engineering involved in integrating and assessing the efficacy of ML models within the clinical workflow is complex. This paper presents a general-purpose, end-to-end, clinically integrated ML model deployment and validation system implemented at UCSF. Engineering and usability challenges and results from 3 use cases are presented. A generalized validation system based on free, open-source software (OSS) was implemented, connecting clinical imaging modalities, the Picture Archiving and Communication System (PACS), and an ML inference server. ML pipelines were implemented in NVIDIA's Clara Deploy framework with results and clinician feedback stored in a customized XNAT instance, separate from the clinical record but linked from within PACS. Prospective clinical validation studies of 3 ML models were conducted, with data routed from multiple clinical imaging modalities and PACS. Completed validation studies provided expert clinical feedback on model performance and usability, plus system reliability and performance metrics. Clinical validation of ML models entails assessing model performance, impact on clinical infrastructure, robustness, and usability. Study results must be easily accessible to participating clinicians but remain outside the clinical record. Building a system that generalizes and scales across multiple ML models takes the concerted effort of software engineers, clinicians, data scientists, and system administrators, and benefits from the use of modular OSS. The present work provides a template for institutions looking to translate and clinically validate ML models in the clinic, together with required resources and expected challenges.

**Funding:** The authors received no specific funding for this work.

**Competing interests:** I have read the journal's policy and the authors of this manuscript have the following competing interests: NVIDIA provided 4 T4 cards as a grant to UCSF.

## Author summary

Academic medical centers gather and store vast quantities of digital data, and with the increase in accessibility of Machine Learning (ML) techniques, there has been an explosion of ML model development in the medical imaging community. Most of this work remains in research, though, and connecting ML models to the clinic for testing on live patient data and integration into the clinical workflow remains a challenge and impedes clinical impact. We present a general-purpose system, implemented and deployed at UCSF, for in-clinic validation of ML models and their incorporation into patient care. This work, based on free and open-source software packages, can serve as a template for other institutions looking to solve ML's "last mile" problem and move their models out of research and into the clinic.

## Introduction

The medical imaging community is embracing Machine Learning (ML) and Artificial Intelligence (AI) to develop novel predictive models. These models show promise, and have the potential to transform radiology practice and patient care, in areas ranging from data acquisition, reconstruction, and quantification, to diagnosis, treatment response, and clinical workflow efficiency [1]. While the foundation of this work is model development using retrospectively acquired datasets [2], translating AI models from research to the clinic for event-driven, prospective validation is a critical step towards model deployment for routine use in clinical care. Prospective model validation within the clinical workflow not only provides an opportunity to capture expert clinical feedback about a model's performance, but is also critical for assessing usability, interpretability, and effectiveness of results, as well as technical issues related to integration with clinical information systems. Moving ML models from "proof-of-concept" to "production" is the critical next-step for medical imaging [3].

The infrastructure and software systems required to clinically integrate models for validation are extensive and can pose major hurdles to ultimately realizing the clinical impact of AI in medicine [4]. Solutions range from commercial products to custom in-house applications [5–8], and offer pros and cons for flexibility, engineering effort, cost, interoperability with clinical systems, support, and usability. The contribution of this paper is a description of a general-purpose end-to-end ML model validation and deployment framework, based on NVIDIA's Clara Medical Imaging [9] software package and the XNAT [10] imaging study management application, that has been developed and deployed at UCSF within The Center for Intelligent Imaging [11] (ci$^2$). The present implementation is built on free, open-source software (OSS) packages and represents a template for other sites wishing to translate and deploy clinically integrated ML models. The overall system architecture is presented together with the benefits and challenges associated with this design, based on experience implementing 3 separate model validation studies, each representing different but commonly occurring clinical use cases in radiology AI.

## Materials and methods

### Data flow

Fig 1 details the end-to-end AI inference system and networks presented in this work. Briefly, DICOM [12] images are sent from scanning modalities at time of acquisition to a DICOM router. The router directs images to the clinical Picture Archiving and Communication System

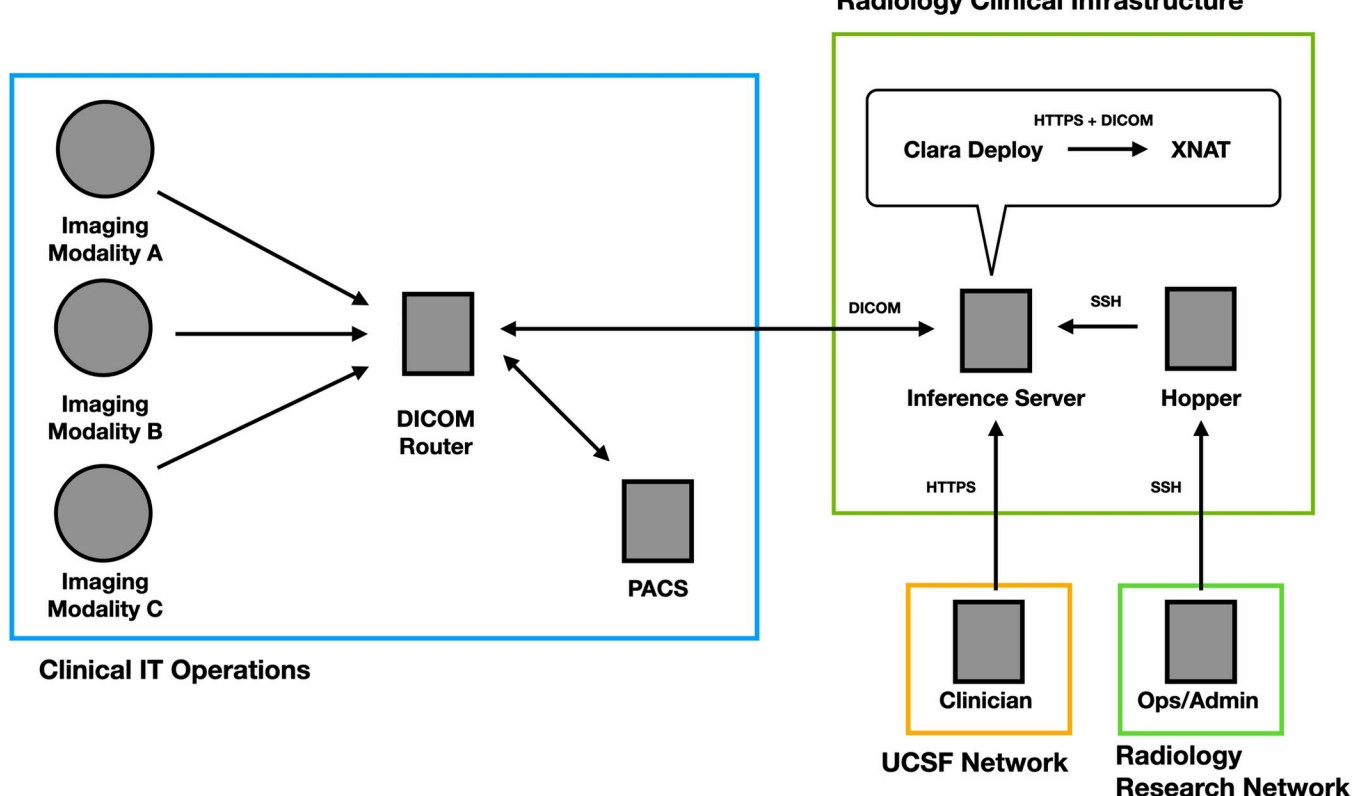

**Fig 1. High-level system architecture and data-flow diagram of on-premises clinically integrated inference validation service.** From top left, clockwise: 1) Clinical imaging modalities and resources networked via DICOM communication protocols. 2) AI inference server running Clara Deploy and XNAT, connected to imaging resources via DICOM. 3) A "hopper" server that gates access to clinical infrastructure, used by system admins and engineers to connect to the inference server from within the radiology network via ssh. 4) Clinicians access the results stored in XNAT from the reading room and other computers within UCSF.

(PACS) [13] and to specific inference services hosted on an on-premises server running NVIDIA Clara Deploy [14], a software platform for deploying ML pipelines. Results are exported to a dedicated instance of the XNAT web application, running on the same host. XNAT stores and displays inference results separate from the clinical record, which clinicians can still access from a PACS workstation in the reading room or other UCSF computers via a browser. Custom buttons in the Visage 7 client (Visage 7 Enterprise Imaging Platform ("Visage 7"), Visage Imaging, San Diego, CA) [15] running on PACS workstations link directly to relevant results in XNAT, where reviewer feedback is captured for use in assessing model performance or for retraining (Figs 2–4).

**Image routing and ingestion.** All DICOM images are sent from clinical scanning modalities to a DICOM router (Compass [16] from Laurel Bridge) that is configured with rules for directing data to various endpoints, including the clinical PACS and the Clara Deploy inference service. Compass' routing rules are a set of user-defined mappings based on DICOM tags in the data. Three rules, corresponding to three proof-of-concept applications, route images to AI inference pipelines (Table 1). Additionally, the inference pipelines are set up as export destinations in the clinical PACS, allowing clinicians to manually transmit images to specific pipelines on-demand. All images are transmitted via DICOM communication protocols.

**AI inference servers.** The system is comprised of both production and development inference servers (Fig 5). These are virtual machines (VMs), hosted on internal UCSF

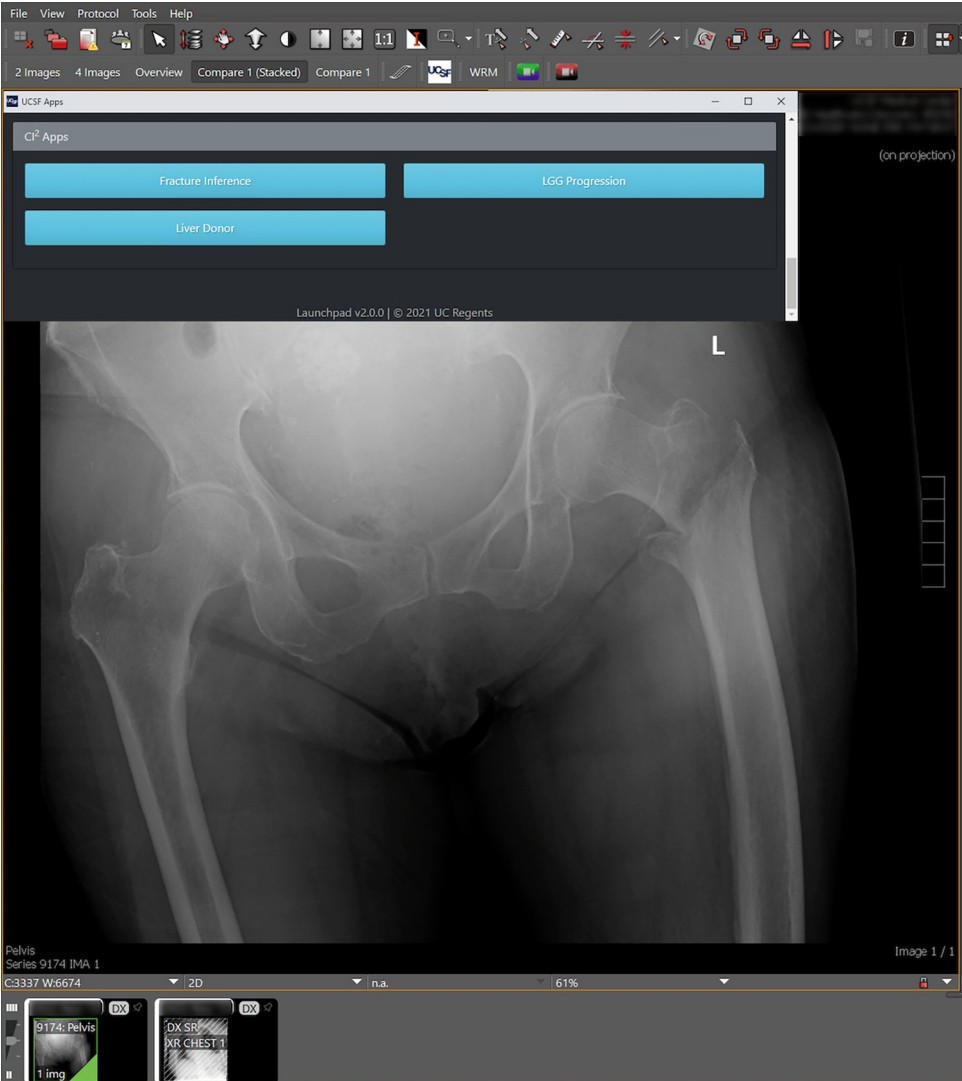

**Fig 2. Workflow for accessing validation studies from the Visage 7 clinical PACS viewer.** Clicking on the custom "UCSF" button in Visage 7 opens a UI panel with buttons linking the current study to ML results and/or the validation study assessment in XNAT.

infrastructure, running on top of VMware's vSphere [17] server virtualization software. The servers run Ubuntu 18.04 [18], and each is assigned a dedicated NVIDIA T4–16c GPU, using NVIDIA Virtual GPU Software's GPU Pass-Through mode [19]. Table 2 summarizes the server infrastructure.

**AI inference framework.** The AI inference pipelines running on these VM's are controlled by and developed in NVIDIA's Clara Medical Imaging application framework. Each VM runs Clara Deploy, a container-based framework for deploying AI workflows. The framework allows developers to build machine learning pipelines that run inference on NVIDIA GPUs, and it supports end-to-end services that include: DICOM import/export, user-extensible pipeline and GPU management, running multiple AI models on GPUs, and interactive image rendering.

Pipelines and services are run in Docker [20] containers and deployed onto the inference server using Kubernetes [21]. AI model inference is run on GPUs using NVIDIA's Triton

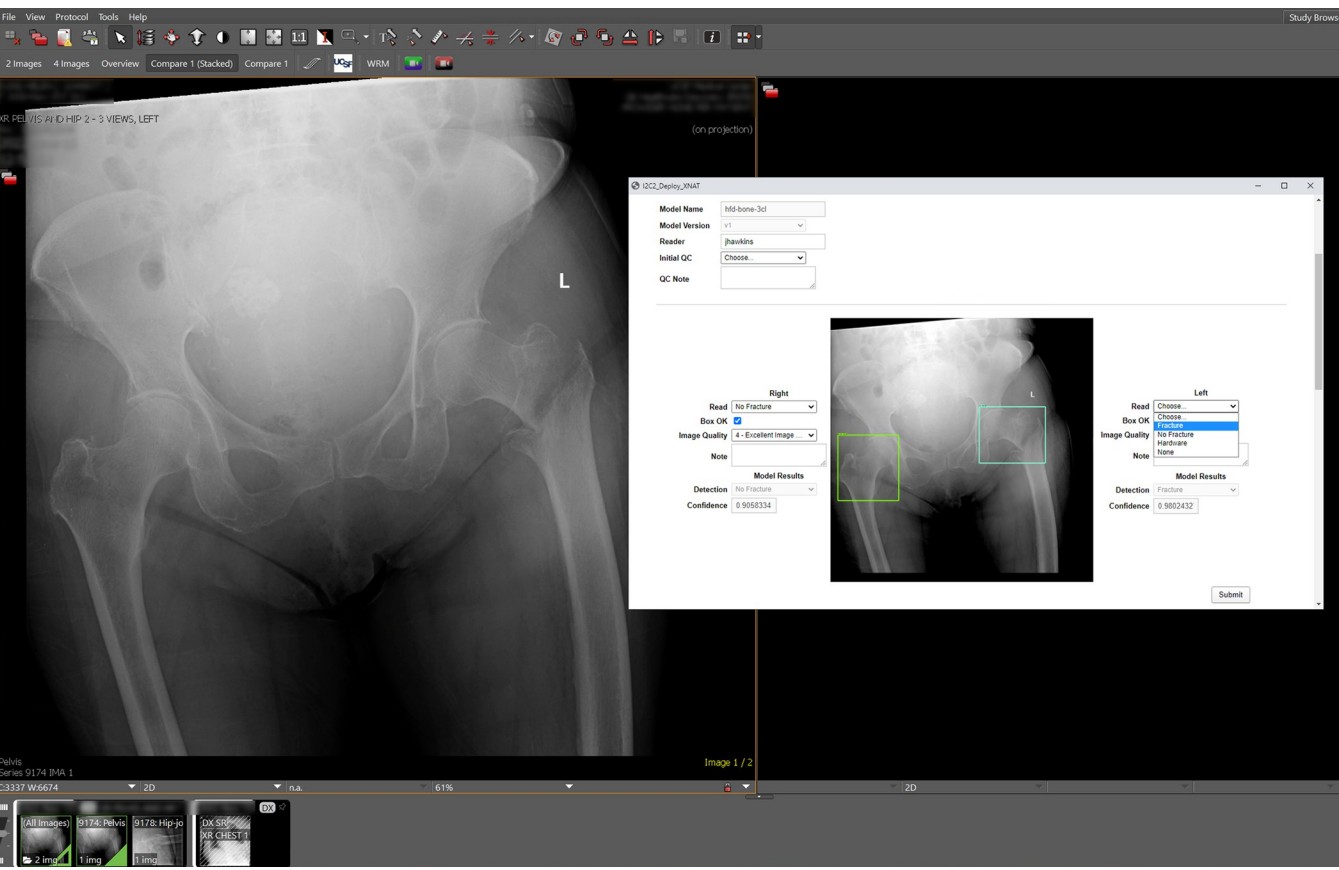

**Fig 3. AI Hip Fracture Detection results and assessment, accessed via Visage 7 link.** When viewing a pelvis x-ray exam in Visage 7, clicking the "Fracture Inference" button (Fig 2) opens a browser window with the exam's Hip Fracture Detection validation study assessment in XNAT, containing: 1) the left and/or right ML model classifications of "Fracture", "No Fracture", or "Hardware" and confidence scores, 2) an image with boxes overlaid where the model has identified the left and/or right hips, and 3) form fields for the reading radiologist to enter their fracture reads and assess the quality of the input image and model results.

Inference Server [22], an AI inference application optimized for GPU performance and bundled with Clara Deploy. The Triton container is managed by Clara Deploy, and is used by pipelines to perform inference, loading models in and out of GPU memory when needed. Pipelines are registered to specific DICOM AE Titles, and when the Clara DICOM Adapter receives a set of images, it looks at the called AE Title and starts processing the images with the associated pipeline.

**Delivery of results.** Inference pipelines export imaging results (e.g., spatial segmentations) as well as scalar classifications and derived metrics to a local, dedicated instance of XNAT, an imaging informatics and study management platform. XNAT was chosen as the mechanism to store and display inference results because of its a) ability to store both imaging and derived scalar data together in one application, b) extensibility, which allows developers to define custom schemas and functionality via its plugin architecture, c) built-in DICOM support, d) security and user permissions model, e) REST Application Programming Interface (API) [23], which allows pipelines to store and modify data via standard web protocols, f) support for the OHIF medical image viewer [24], and g) customizable web-based user interface, which can be tailored to meet the data visualization, feedback capture, and workflow requirements of each inference pipeline. Using XNAT as a data store is a key component of this

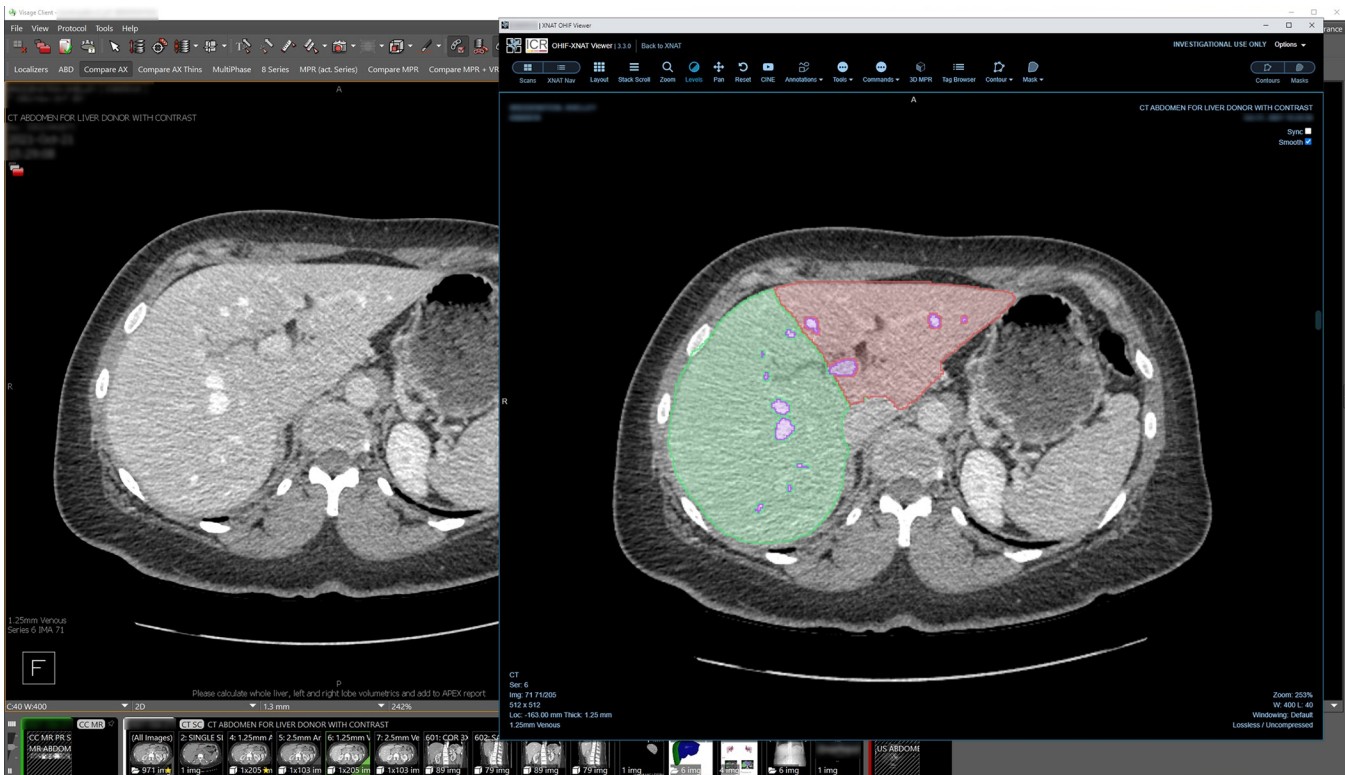

**Fig 4. AI segmentation of the liver for the Liver Transplant Segmentation validation study, displayed in XNAT's image viewer, via Visage 7 link.** The "Liver Donor" button in Visage 7 (Fig 2) links a liver CT exam with its Liver Transplant Segmentation validation study results in XNAT. Using XNAT's image viewer, a clinician can view and edit the ML model generated segmentations of "Left Lobe," "Right Lobe," and "Vessels".

system, as validation study results are separate from the clinical record but can still be integrated within a clinician's workflow.

The XNAT application and its Postgres [25] database are each run inside Docker containers deployed on the inference server, and HTTPS communication is proxied through an NGINX [26] container. XNAT user accounts and authentication are integrated with UCSF's Active Directory Service [27], via the LDAP [28] protocol.

**Table 1. Compass routing rules for the 3 AI inference pipelines described in this work.**

| Clara Pipeline Destination | Called AE Title | DICOM Tag Name | DICOM Tag Group/Element | DICOM Tag Value |
|---|---|---|---|---|
| **Brain Tumor Segmentation** | CI2_CD_BTS | SOP Class UID | (0008,0016) | MR Image Storage |
| | | Study Description | (0008,1030) | "BRAIN*NAV" |
| | | Station Name | (0008,1010) | A set of pilot scanners |
| **Liver Transplant Segmentation** | CI2_CD_LDN | SOP Class UID | (0008,0016) | CT Image Storage |
| | | Study Description | (0008,1030) | "ABDOMEN FOR LIVER DONOR WITH CONTRAST" |
| | | Station Name | (0008,1010) | A set of pilot scanners |
| **Hip Fracture Detection** | CI2_CD_HIP_FRAC | SOP Class UID | (0008,0016) | CR Image Storage |
| | | Study Description | (0008,1030) | "PELVIS" |
| | | Station Name | (0008,1010) | A set of pilot scanners |

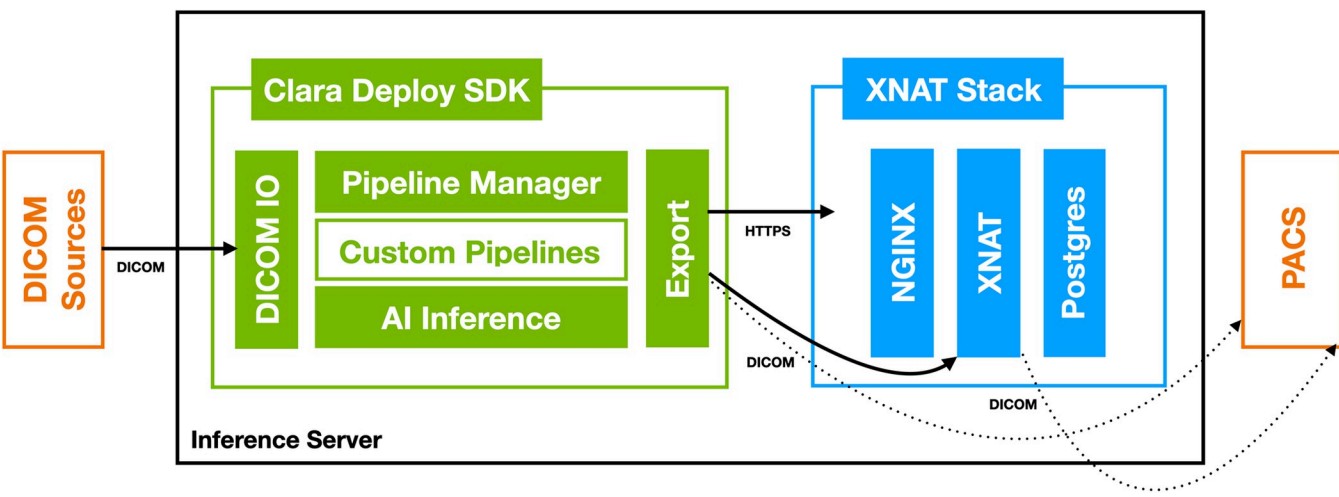

**Fig 5. Inference VM internal architecture diagram.** Green denotes Clara Deploy's backend components, and blue XNAT's.

## Inference pipelines

AI inference pipelines need to perform a consistent set of tasks. The first step often involves parsing an imaging exam to find the relevant input series. Next, images typically require pre-processing, such as intensity normalization, cropping, resampling, and/or registration. Following inference, additional post-processing operations may be required, for example computing derived metrics such as segmentation volumes. Finally, results must be exported to a data management system such as XNAT or a PACS. In Clara Deploy, each of these tasks are implemented as independent software units, called "operators". Pipelines are composed of series of chained operators, each running as a Docker container. Each operator receives data from its preceding operator, via shared data mounts, and performs one processing task before passing output to the next operator. This architecture allows for the reuse of general-purpose operators and extensibility of other algorithm modules for new pipelines. Fig 6 illustrates a typical image segmentation pipeline archetype.

**NVIDIA Clara Deploy operators.** NVIDIA provides a library [29] of Clara Deploy operators as Docker images that can be used to compose pipelines, including operators for DICOM reading/writing, exam parsing, series selection, and deployment of Clara Train developed AI models. NVIDIA also provides a base Docker image which can be used to develop custom operators for additional functionality and integration of models developed outside of the Clara Train framework. The pipelines in this work use a mixture of NVIDIA's standard Clara Deploy operators, modified operators that extend standard Clara Deploy operators, and fully in-house developed operators (Table 3).

**Table 2. Inference infrastructure configuration.**

| Physical Server | Cisco UCS C240-M5s |
| --- | --- |
| | 2 x NVIDIA T4 GPUs |
| | NVIDIA GRID software version 11.1 (installed on ESXi host) |
| **Virtual Machines** | 8 vCPUs (Intel Xeon Platinum 8168 CPU @ 2.70GHz) |
| | 32G RAM (reserved) |
| | PCI Device: NVIDIA GRID vGPU |
| | GPU Profile: grid_t4–16c |

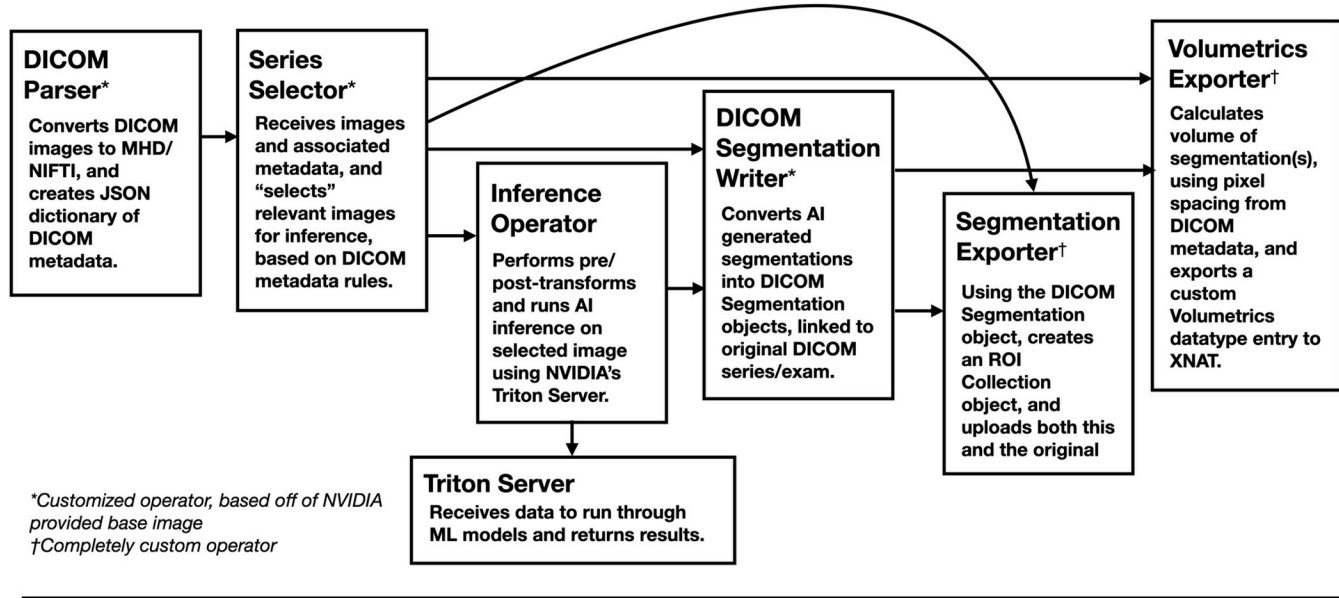

**Fig 6. Anatomy of an ML inference pipeline.** Each box represents a pipeline operator executing a Docker container, managed by Clara Deploy.

Modified Clara Deploy operators are derived from reference Clara Deploy operators, where the application source code has been customized by ci² engineers to fit specific pipeline needs. Modifications were necessary, for example to handle data in unexpected formats, to support

**Table 3. List of Clara Deploy pipeline operators used, with estimate of software engineering time necessary to develop similar functionality.**

| Operator Name | Purpose | Type | Effort |
|---|---|---|---|
| Clara DICOM Reader | Ingests a DICOM series and converts to MHD [30] /NIfTI [31] | Included | |
| Clara Deploy DICOM Parser | Ingests a DICOM exam and parses DICOM metadata for use with the Series Selector | Included | |
| Clara Deploy Base Inference V2 | Performs AI model inference with Clara Train developed models | Included | |
| Clara Register Results | Used to export results to DICOM receivers | Included | |
| Clara DICOM Writer | Writes AI model results into DICOM format; modified to allow for custom DICOM tag values | Included | |
| Clara Deploy DICOM Parser | Modified to add in additional DICOM tags to the parsed metadata, to parse/convert x-ray exams with multiple instances per series, and to continue converting series in an exam if there is a failure converting a series with unexpected DICOM attributes | Modified | ~1 day |
| Clara Deploy Series Selector | Modified to add in regular expression parsing, to move selected series to output directories, and provide the option to select individual x-rays on an instance (non-series) level | Modified | ~1 day |
| Clara Deploy DICOM Segmentation Writer | Developed in parallel with NVIDIA, and modified with the ability to customize DICOM tag values | Modified | ~1 week |
| DICOM RTSTRUCT Writer | Modified to handle image ordering and orientation overlay issues, and add the ability to customize DICOM tag values | Modified | ~1 week |
| Hip Fracture Detection Inference Operator | Used to deploy a non-Clara Train image classification model (TensorFlow [32] Object Detection [33] based) | In-House | ~1 week |
| Clara Deploy XNAT ROI Collection Exporter | Used to export segmentation results to XNAT, and create segmentation feedback entries in XNAT | In-House | ~1 week |
| Clara Deploy Volume Calculator | Used to calculate segmentation volumes and export them to XNAT. | In-House | ~1 week |

Efforts listed are based on estimates for an experienced software developer familiar with Python, medical imaging APIs and containerization technologies, and will vary based on skill level and experience with requirements, underlying technologies, and interfaces.

pipeline specific selection of a subset of exam data relevant for inference, or to read/write DICOM metadata needed for clinical integration. These modifications are possible because the Clara operator source code is open and accessible, removing the need to rely on NVIDIA engineers to implement feature requests.

Entirely new operators, written in Python [34], are developed to provide functionality not present in Clara or integrate non-Clara Train based models. The application code is built on top of a base Docker image containing the Clara libraries necessary to integrate with the Clara Deploy framework and other pipeline operators. NVIDIA provides a downloadable Operator Development Kit [35], which walks through building Clara Deploy operators from scratch. It includes a sample segmentation model, imaging data, and the source code to build and run a custom inference operator.

## Management of results

Inference results during validation are stored separately from the clinical record, using XNAT, which provides a Java [36] plugin architecture [37] to support custom data schemas and extend functionality. Custom plugins were built for each pipeline to store non-DICOM output (such as volumetrics and classification results) with the corresponding images, customize how results are displayed to clinicians, and define feedback forms to assess model performance and clinical efficacy. Each validation study has a custom feedback capture form integrated with the results, with freeform text inputs, dropdowns, and other standard HTML inputs. Clinicians can make non-destructive edits to ML segmentations, with edited segmentations saved back into XNAT and linked to the source exam. The AI model name and version used to generate the results are stored with all results. Table 4 lists the plugins used in the present system and indicates which were developed in-house.

A sample XNAT results page from the Brain Tumor Segmentation pipeline (described below) is shown in Fig 7. When a clinician views this page, they can see: the calculated tumor volumes for the patient's current and prior exams (in table and graph format); the percent change of each tumor volume, relative to a baseline, and whether that percent change is above a threshold for tumor progression; the segmentation overlaid on top of the source DICOM image; and a feedback form. Clinicians can view the segmentation results, assess tumor

**Table 4. List of XNAT plugins, with estimate of software engineering time necessary to develop similar functionality.**

| Plugin Name | Purpose | Development | Effort |
|---|---|---|---|
| XNAT-OHIF Viewer Plugin | DICOM image viewer, with segmentation and ROI contour support [38] | XNAT Team | |
| XNAT LDAP Authentication Provider Plugin | Integrates XNAT user accounts with UCSF's Active Directory system for authentication [39] | XNAT Team | |
| XNAT Container Service | Controls processing jobs using Docker containers on data stored in XNAT [40] | XNAT Team | |
| Hip Fracture Datatype | Stores inference results and feedback | In-House | ~1 week |
| Brain Tumor Segmentation Datatype | Stores inference results and feedback | In-House | ~1 week |
| Liver Donor Segmentation Datatype | Stores inference results and feedback | In-House | ~1 week |
| ROI Volume Datatype | Stores segmentation volume measurements, linked to the segmentation, model name, and model version | In-House | ~ 1 day |
| DICOM Import Identifier | By default, XNAT pulls subject and session information from the DICOM tags PatientName and PatientID when storing images; To integrate with clinical data, this plugin sets up the DICOM SCP to use the DICOM PatientID and AccessionNumber tags to define subject/session | In-House | ~1 day |

Effort will depend on skill level and experience with underlying technologies and interfaces.

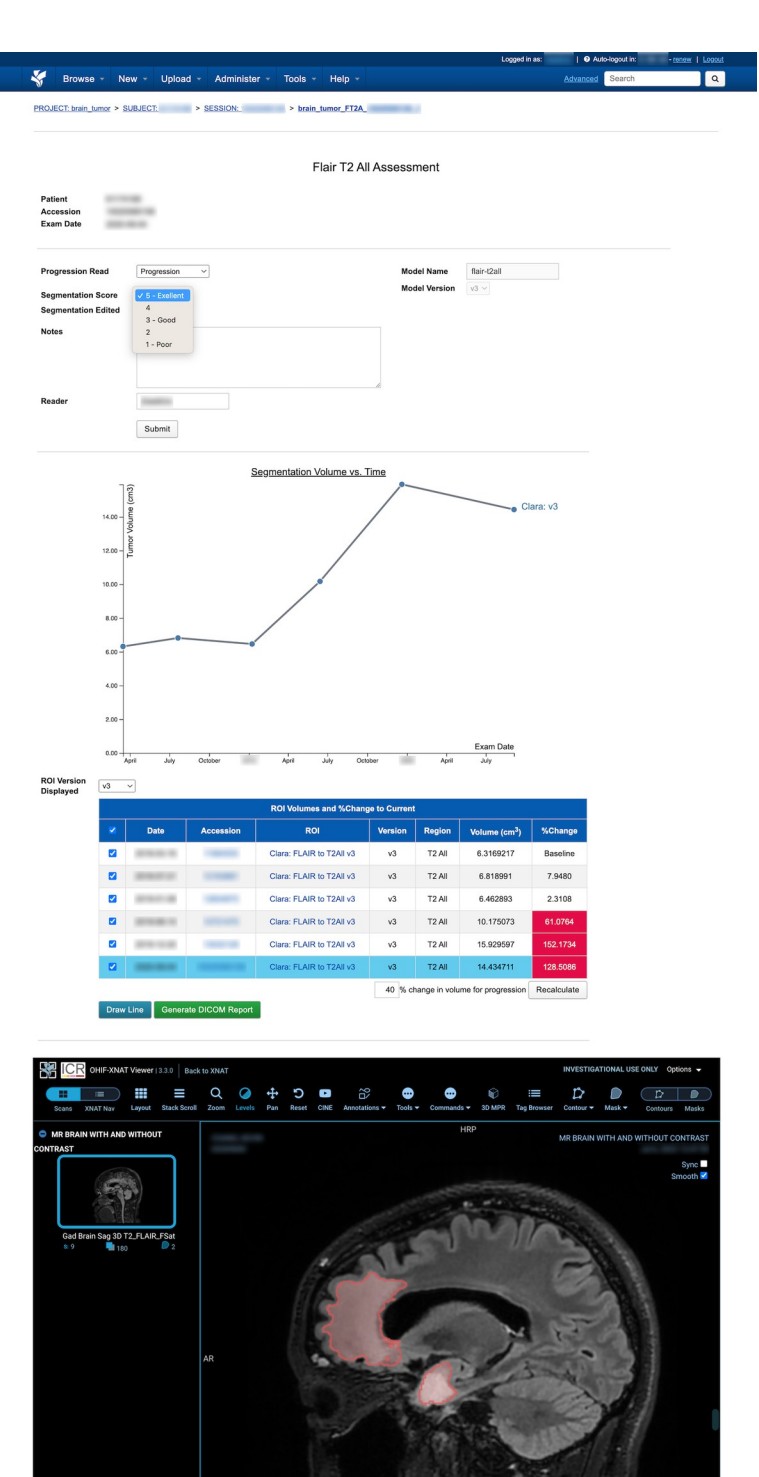

**Fig 7. Brain Tumor ML Segmentation and progression display with feedback capture webpage in XNAT.** Contains form for assessing model results (top), interactive plot and table showing the patient's segmentation volumetrics at each exam time point (middle), and embedded OHIF viewer with editable lesion segmentation (bottom). The data table and plot display the volumes of the ML model segmentations and/or the manually edited segmentations. In the data table, red cells represent progression according to a user-defined threshold of a 40% increase in volume over baseline. A clinician can generate a DICOM Secondary Capture report containing the plot and table by clicking the "Generate DICOM Report" button.

progression, and leave feedback on model performance. They can also edit the model's segmentation and save a corrected copy back into XNAT. The volume of an edited segmentation is automatically calculated, via XNAT's container service [40] and an in-house developed volume calculation container, and added back into the displayed tumor volume table and plot (without overwriting the original inference results).

## Pipeline development and deployment

Operators, pipelines, and XNAT plugins are developed and initially deployed on a clinical VM dedicated to testing. This test VM runs its own instances of Clara Deploy and XNAT. Pipeline definitions and XNAT plugins/configurations are pulled from an on-premises Gitlab [41] instance and deployed. Kubernetes and the Docker daemon are configured to pull Docker images from the Gitlab container registry. Test data cohorts are manually sent through pipelines via DICOM transfers from PACS, and clinicians review results in XNAT for feedback on usability, design, and the metrics that should be captured about AI model performance. Once it's verified that a pipeline can successfully ingest a clinical exam, select the correct image/s for processing, and output the expected inference results, the inference pipeline is deployed on the production VM in the same manner.

As these inference services are integrated with clinical resources, they reside on infrastructure maintained by Radiology Clinical IT. This provides a high level of monitoring and service support, which is necessary for the service up-time required by clinicians participating in validation studies, including after-hours. This also provides an added layer of security, as the inference VM's are isolated behind clinical firewalls. As a result, only authorized personnel have access to these systems for deployment and operations. Per UCSF policy, this system has undergone an IT Security Risk Assessment, which reviews all IT projects for potential data and security risks.

## Proof of concept validation projects

Three AI models, described below, were chosen to pilot clinical pipeline integration. Two of the models were developed in Clara Train, using built-in model architectures, and trained on imaging data acquired at UCSF. The third model was trained and developed outside of the Clara Train framework, utilizing TensorFlow's Object Detection API [33]. The 3 clinical validation studies involved clinicians from different departments within UCSF (Radiology and Biomedical Imaging, Surgery, and Emergency Departments), and received institutional review board approvals with consent waivers. The purpose of these proof-of-concept (POC) projects in the present work is to assess model deployment and integration. Model development and training is beyond the present scope and details are provided in references below.

**Brain tumor segmentation.** A Clara Train 3D U-Net [42] was trained to segment non-enhancing lesions from 3D post-surgical MRIs of patients with low grade gliomas (LGG). Segmented volumes are used to compute tumor volume for the current exam and priors. This was incorporated into a deployed pipeline that aimed at detecting volumetric changes from baseline to monitor for disease progression. A clinical validation study was run to assess whether

AI-based segmentation could be incorporated into patient care to detect non-enhancing glioma progression (Fig 7) [43].

**Liver transplant segmentation.** The same Clara Train 3D U-Net model architecture was used to develop a liver segmentation model for use in surgical planning for transplants [44]. A pipeline was developed to automatically segment both the left and right liver lobes as well as vessels from CT images and then calculate volumetrics (Fig 4). Surgeons use the segmentations and the calculated volumes to determine transplant viability and plan the surgical approach. This project captures timing metrics, of both segmentation pipeline execution and review of the results, to compare against current manual and semi-automated segmentation workflows.

**Hip fracture detection.** A third pipeline utilizes an object detection and classification model to localize the left and right hip in x-ray images, and classify each as normal, containing a fracture, or as having surgically implanted hardware [45]. This model was developed by UCSF's Musculoskeletal Quantitative Imaging Research group [46], outside of Clara Train, using the TensorFlow Object Detection model framework. Compass is configured to route pelvis exams from 2 x-ray scanners in the UCSF Emergency Department to the Clara Deploy inference server (Table 1). The deployment is being assessed for its ability to improve emergency room outcomes by improving hip fracture diagnosis and reducing a patient's time to treatment (Fig 3).

## Results

The system detailed above was used to deploy the 3 POC projects to support validation studies aimed at characterizing all aspects of pipeline development and integration from data flow to system performance, extensibility, engineering robustness and usability. The present section focuses on results related to characterizing the system's viability as a general-purpose platform for supporting clinical validation of AI models for a variety of representative workflows, workloads and use cases. Specific details pertinent to the clinical use, model performance, and clinical impact of each model is beyond the scope of this paper and will be presented in separate papers.

The Brain Tumor Segmentation (BTS) pipeline initially received 30–40 exams per week via automatic Compass routing, from 2 clinical MRI scanners (Table 1), for inference. The imaging protocols that incorporate the sequence used to train the model last around 40 minutes, and images were routed to our Clara VM over that entire timeframe, with a typical exam containing about 1GB of data. The BTS pipeline, which segments LGG tumor and calculates the segmentation volume, takes on average 2.9 minutes to execute per exam, including the time to transfer the results to XNAT, with 90% of cases processing in under 4.5 minutes. This execution time is not only sufficient for processing automatically routed cases, but also met the requirements of radiologists participating in the model validation study, who requested a <10-minute turnaround time per exam when manually transmitting images from PACS. A longitudinal analysis to assess tumor progression for a patient over 6 timepoints takes <18 minutes to process. A clinical neuroradiologist reviewer, logged-in to a PACS client, was able to search for, transmit, and receive results for 65 current and retrospective MRI exams, across 10 LGG patients, in a 3.1-hour window. The segmentation pipeline completed successfully for all cases. Results and findings from this validation study were reviewed by ci[2]'s Clinical Deployment committee (see Discussion: Governance and Validation Criteria), which supported adoption of the pipeline for routine clinical use at UCSF.

The Liver Transplant Segmentation (LTS) pipelines received 3–5 exams per week via automatic Compass routing. This study involved blind reading of 3 different segmentation results from the same exam: expert human reviewer, novice human reviewer and machine

segmentation. Human reviewers require 1–2 hours to produce the segmentation, whereas AI results can be delivered in less than 10 minutes. Each segmentation is identified with a unique salted hash that is inserted into the series description of the DICOM Segmentation Object when it is written. This identifier is than stored as a text file which is passed to the ROI Collection Exporter and the Volume Calculator (Table 3) to ensure that the source of the segmentation is retained but appropriately obscured from the reader. Three different clinicians then reviewed the segmentation in XNAT via OHIF and provided feedback in forms linked with each case's unique hash. Clinicians reported review times of 1–5 minutes per case.

Over 13 weeks, the Hip Fracture Detection (HFD) pipeline processed 200 exams from 1 emergency department x-ray scanner, sent automatically via Compass routing. Exams typically contained five 2D images of 5MB each. The observed transfer time for a single exam was <1 minute, which defined the patient level time-out for triggering the pipeline, and the inference pipeline's average run time was 20 seconds, including uploading of results to XNAT. The workflow for this pilot study incorporated a "human in the loop" (HITL) step, where a member of the ci$^2$'s 3DLab [47] assessed the quality and relevance of each input x-ray image that was processed, before placing the inference results on the XNAT worklists of the 2 participating clinical readers, who assessed whether the AI model correctly identified the hip joints in the image and made their own read on whether each joint contained a fracture, no-fracture, or hardware. The HITL quality control workflow was implemented in the HFD XNAT plugin and took the reviewer 1 minute per exam.

During the 2 years of initial testing, validation study ramp-up, and the current clinical deployment of the BTS pipeline, the inference system has segmented more than 100 liver ROIs, processed over 600 LGG exams, and run fracture-detection on 2000+ pelvic x-ray studies. On average, it is running automatic inference on 5 hip exams per day and 25 LGG exams on-demand per week, without negatively impacting clinical infrastructure or needing technical support beyond standard maintenance and functionality updates.

## Discussion

Deploying and supporting an ML pipeline in the present framework requires software development and system engineering on multiple fronts. The model must be trained, AI inference operator built, and pipeline execution steps designed; pipeline operators performing additional calculations or data tasks must be built; XNAT plugins need to be developed to store and display result and capture user feedback; finally, data ingestion, pipeline execution, and results display must be tested with clinical data, which will differ from research data in unforeseen ways. Operator and plugin development efforts are estimated in Tables 3–4, but will vary based on skill level and experience. Collaboration with the clinical users is critical to define data display and data flow requirements. At UCSF, ci$^2$'s Computational Core [48] supports this effort bridging the gap between scientific research, software engineering and enabling translation of AI research into the clinic.

### Building an open-source based inference system

The choice to utilize a commercial ML deployment product or build one in-house will depend on the degree of flexibility and customization required. Licensing costs and implementation time must be considered, but the need to integrate with a specific clinical workflow or proprietary endpoint may point towards a commercial product. Commercial solutions often require writing source code against private APIs and building Docker containers for inference. Additionally, deployments rely on vendor supported integrations and visualizations. In the present case, the need for a general-purpose and self-customizable framework capable of supporting a

broad range of use-cases, data presentations that integrate a range data types, feedback capture, and integrations with multiple clinical modalities and information systems led to an in-house solution based on an open-source software ecosystem.

**Deploying on-premises.**   When setting up software systems, organizations are faced with the choice of deploying to on-premises hardware, externally hosted cloud infrastructure, or a hybrid of the two. A clinical system processing and storing Protected Health Information (PHI) must comply with institutional policies, which dictate how data can be stored and whether it can leave internal infrastructure. HIPAA Business Associate Agreements (BAAs) can be established with cloud providers to allow for PHI data sharing, but this can be a lengthy process with an institution's legal and IT security teams. For this project, leveraging cloud-based infrastructure was not a viable option, but policy aside, there are several factors that make on-premises deployment a good choice for this system.

Cloud computing provides value by abstracting away hardware provisioning and systems administration. At UCSF, the Clinical Infrastructure team and its resources can be leveraged to stand-up and manage an inference server tailored to the needs of this project. By building on internal resources, the presented infrastructure inherits monitoring and security tools that already support existing clinical applications. Data routing and security becomes simpler, as all traffic is routed internally and on existing networks, using already approved data flow channels set up to communicate with clinical resources. At current workloads, the cloud's ability to quickly scale resources on-demand is not necessary, though owning and controlling the present software means deploying to additional on-premises servers or the cloud is possible if needed in the future.

**Software vulnerabilities and system security.**   While there is academic debate over the security risks and benefits of open versus closed source software [49,50], the OSS stack underpinning this deployment has been compliant with UCSF's IT Security framework. The host undergoes weekly security scans by Tenable's Nessus [51] vulnerability assessment platform. Ubuntu security patches are applied by a Clinical Infrastructure systems administrator, and application-level vulnerabilities found within the deployed containers are reported to ci$^2$ engineers for remediation. This involves rebuilding application containers with up-to-date base Docker images and dependencies, and updating the configurations on the inference server to pull them. As a best-practice, the latest base Docker images are pulled and full builds are run to refresh dependencies when pipeline operators or XNAT are updated. During the 2 years since the system's initial deployment, no critical vulnerability has caused downtime.

## Pipeline development efforts

Deploying the Clara Deploy framework and linking it with imaging sources requires knowledge of DICOM protocols, tooling, experience with Python application development and containerization. Development requires familiarity with multiple imaging data formats, AI development frameworks and communication with web services via REST API's. The use of XNAT to create interactive data views entails web development skills and since the framework integrates with clinical systems, knowledge of security best-practices is critical.

**Pipeline deployment considerations.**   Clinical Integration of AI pipelines involves collaboration across multiple organizational units. ci$^2$ engineers coordinate with: Clinical IT, to configure image routing and PACS integration; Clinical Infrastructure, which hosts and maintains the VM's and networking; data scientists and researchers who develop AI models; and clinicians, to define image routing rules, develop effective visualizations in XNAT, gather model feedback, and determine how AI results can integrate with already complex clinical workflows. Models are increasingly incorporating imaging and non-imaging data, e.g., from Electronic

Medical Records (EMRs), further increasing the complexity of the landscape. UCSF's APeX Enabled Research (AER) group [52] provides a support path for SMART-on-FHIR [53] enabled EMR access for translational work.

**Deployment framework flexibility.** Academic medical centers have diverse sets of research groups, spread across departments, doing ML model development. Groups will have independently developed unique model training toolsets, using custom software based on a variety of ML frameworks. A deployment system needs to support integrating models and their supporting code from outside of its ecosystem. While Clara Deploy supports running Clara Train developed models by building a configuration file into the base inference operator, it was also possible to integrate the HFD TensorFlow Object Detection model into a pipeline by building a custom inference operator with refactored research code and the Clara APIs. Clara Deploy supports running inference with models from external frameworks, provided inference can be run within a Python executable. Engineering teams supporting clinical ML deployment need to encourage scientific research groups to follow software best-practices, as integrating research ML models into reliable clinical pipelines requires software to be packaged into documented, reusable libraries.

**Deployment framework modularity.** Deploying our first pipeline (BTS) required modifying 3 Clara Deploy operators and the development of 2 custom operators (see Fig 6 and Table 3). The LDS pipeline was able to re-use all of those operators, significantly reducing the engineering effort to deploy. Many pipelines have similar pre- and post-inference needs, and scaling functionality across use-cases is integral to supporting the deployment of multiple ML models.

**Deployment framework updates and transitions.** Leveraging established, well-tested and supported third-party software frameworks for development offers significant advantages for development cost and product stability; it does, however, pose risks that may include managing changes to APIs or dependencies losing maintenance support and which would present substantial implications for project effort and direction. Choosing to work with open-source software that has strong, communicative leadership is key to mitigating such risks. Building on open standards and industry protocols [54] ensures code portability, and communication within a framework's community will lead to smoother upgrade cycles. NVIDIA's Clara project is transitioning [55] into the Medical Open Network for Artificial Intelligence (MONAI) [56], and though work will be necessary to move from Clara Deploy to MONAI Deploy, the 2 projects' open natures and strong communication within the MONAI Working Group (which includes NVIDIA) [57] promise a minimally impactful transition.

## Access

As the primary purpose of the clinical PACS or EMR system is to directly support patient care by providing physicians access to data, clinical IT teams must prioritize the stability and performance of clinical use cases. Any new system that plugs into infrastructure critical to patient care is a potential risk, must be robust and not negatively impact network infrastructure or IT support teams. Clinical integration of a translational framework may thus require flexibility to adapt to authorized access methodologies.

Another design consideration during a validation study is balancing the need to segregate AI validation data from the clinical record while simultaneously providing seamless access to AI results for clinical readers within their existing workflow. The present XNAT-based approach to storing AI-derived results is thus designed to facilitate prospective clinical validation of AI models, by providing clinicians seamless access to results from a button in the clinical PACS, while validation results and feedback are stored separately from clinical information

systems (Figs 2–4). A web-based approach allows clinicians flexibility in how and when they review cases, but does not necessarily represent a final solution that fits into a clinical workflow.

## Clinical workflows

Radiologists are faced with demanding workloads, and validation workflows must be designed with efficiency in mind, as every additional mouse-click represents an obstacle to adoption. [58] Any new information that an AI model provides must yield clear, concrete improvements to patient care or a clinician's workflow. Ideally, new information would be integrated into an existing tool; however, radiology workflows are largely built around commercial applications which may or may not offer endpoints or APIs for an AI pipeline to interact with. Moreover, even when integration points to hospital wide applications (PACS, EMR) exist, obtaining access approval for translational work may entail a lengthy and uncertain approval process. AI results should be stored in standardized, open formats to allow for flexibility in presentation method within the clinical workflow [59].

## Governance and validation criteria

The decision to use AI results in routine patient care and include them in the clinical record requires careful consideration and a defined governance plan [60]. ci²'s Clinical Deployment committee [61] provides governance over such decisions and reviews all potential AI applications through a structured cost-benefit analysis process. Application evaluation considers model accuracy, connectivity, and robustness, and the potential impacts to operations and workflow. Pipelines must seamlessly deliver consistently interpretable results within a clinical context. Operational cost, mode of integration, and the benefit and risk to patients and clinicians for reasons ranging from potential model bias to implications from erroneous results and physician "automation complacency" [62] are considered. Ultimately, a pipeline needs to provide clear improvements over the existing standard-of-care.

The current platform provides a streamlined mechanism for gathering the real-world feasibility and performance metrics necessary for a governance body to assess whether a given model and implementation is a candidate for routine clinical use.

## Conclusion

Implementing a generalized, extensible, and scalable platform for validating and deploying AI-based pipelines in the clinic takes time and effort from a dedicated engineering team, in collaboration with clinical end users capable of providing guidance on usability and requirements. There is a considerable amount of work in system design, infrastructure setup, and software engineering to ensure high reliability and support for a diverse set of workloads and workflows, but the upfront investment does return significant value. The server and network architecture put in place is positioned to adapt to support new standards-compliant ML model deployment frameworks used down the road, should a software transition become necessary, and once connectivity with clinical systems is operational the same architecture will support additional servers and pipelines.

Similarly, choosing to use a deployment framework designed to be modular and built on open-source tooling will have benefits for the future. Clara Deploy's modularity has meant that after functionality has been developed for one pipeline, it can be re-used in future workflows, dramatically decreasing the time to deploy new AI models that share similar pre- and post-processing needs. The ability to access and extend Clara Deploy operator source code was essential to developing pipelines and operators that can interact with clinical data and

resources that always have edge cases that differ from what a framework's developers expect. Building on standard, open-source software tools also ensures a level of portability should deployment requirements or frameworks evolve.

The inclusion of XNAT to store results external to the clinical record was also key in developing an AI model validation framework. XNAT is valuable not only as a multi-modal data manager, but also for its extensibility, which allows it to act as a customizable validation study platform. Having a web frontend to AI results also enables rapid iteration on interactive UIs for presenting model output and generating final reports on findings.

While commercial ML model deployment options exist, the choice to build an in-house solution preserves flexibility in data routing, infrastructure, ML model framework choice, and project-specific workflow, visualization, and validation requirements. This is particularly important for supporting translational work for a broad range of use-cases being developed in a large research institution. Leveraging robust open-source components significantly reduces development efforts while providing adaptability and improving resilience. The process and systems outlined above have been demonstrated to provide a flexible and dependable ML model deployment platform, that will scale across pipelines and use-case specific requirements and handle the deployment process from validation study to integration into the clinical workflow.

## Acknowledgments

For their hard work in supporting the infrastructure and networking in the above system, we would like to thank: Jeff Block, Matt Denton, and Reese Webb from Radiology Clinical Infrastructure; Peter Storey and Jed Chan from Radiology Scientific Computing Services; Neil Singh and Muse Hsieh from Radiology Clinical IT Operations; and Dr. Wyatt Tellis from Radiology Innovation and Analytics. We would also like to thank Dr. Mona Flores, Dr. Sidney Bryson, Rahul Choudhury, Victor Chang, and David Bericat from NVIDIA for help deploying and developing Clara at UCSF.

## Author Contributions

**Conceptualization:** James R. Hawkins, Marram P. Olson, Christopher P. Hess, Sharmila Majumdar, Jason C. Crane.

**Methodology:** James R. Hawkins, Marram P. Olson, Ahmed Harouni, Ming Melvin Qin, Jason C. Crane.

**Software:** James R. Hawkins, Marram P. Olson, Ahmed Harouni, Ming Melvin Qin.

**Writing – original draft:** James R. Hawkins.

**Writing – review & editing:** James R. Hawkins, Marram P. Olson, Christopher P. Hess, Sharmila Majumdar, Jason C. Crane.

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
