## [Decision Letter · Decision Letter 0]

25 Apr 2023

PDIG-D-23-00075

Implementation and prospective real-time evaluation of a generalized system for in-clinic deployment and validation of machine learning models in radiology

PLOS Digital Health

Dear Dr. Hawkins,

Thank you for submitting your manuscript to PLOS Digital Health. After careful consideration, we feel that it has merit but does not fully meet PLOS Digital Health's publication criteria as it currently stands. Therefore, we invite you to submit a revised version of the manuscript that addresses the points raised during the review process.

Please submit your revised manuscript within 30 days May 25 2023 11:59PM. If you will need more time than this to complete your revisions, please reply to this message or contact the journal office at digitalhealth@plos.org. Please include the following items when submitting your revised manuscript:

We look forward to receiving your revised manuscript.

Kind regards,

Yuan Lai, Ph.D.

Academic Editor

PLOS Digital Health

Journal Requirements:

Additional Editor Comments (if provided):

Reviewers' comments:

Reviewer's Responses to Questions

**Comments to the Author**

1. Does this manuscript meet PLOS Digital Health’s publication criteria? Is the manuscript technically sound, and do the data support the conclusions? The manuscript must describe methodologically and ethically rigorous research with conclusions that are appropriately drawn based on the data presented.

Reviewer #1: Yes

Reviewer #2: Yes

2. Has the statistical analysis been performed appropriately and rigorously?

Reviewer #1: N/A

Reviewer #2: N/A

3. Have the authors made all data underlying the findings in their manuscript fully available (please refer to the Data Availability Statement at the start of the manuscript PDF file)?

Reviewer #1: Yes

Reviewer #2: Yes

4. Is the manuscript presented in an intelligible fashion and written in standard English?

Reviewer #1: Yes

Reviewer #2: Yes

5. Review Comments to the Author

Reviewer #1: I commend the authors for creating a well explained report on the implementation of a ML pipeline into clinic workflow. This paper discusses the infrastructure developed and tested for deploying ML model validation and deployment using open source software. This systems level work may be used as template for other institutions that are interested in utilizing ML capabilities for medical segmentation and or classification tasks. 

General review 

The manuscript is well written and easy to understand. The figures are are descriptive and the tables well enumerate the difference between plug and play software vs. in-house development of software. The system was well tested. 

My concern is that with open source software the issues become apparent only with the passing of time. The risks of software becoming outdated or “orphaned” software, untracked dependencies etc. may pose a risk down the line. Often it can take up to a year for vulnerabilities to show up in open source based software. 

Is there a DevSecOp built into the process? and does the cost -benefit analysis account for the time/expertise to keep the pipeline secure/stable. I would urge the authors to write a short status report in 18 months timeframe to report on vulnerabilities(if any). 

Review by section 

Materials and methods -This section is clear and well explained. My only suggestion for this section is to release some in-house development code via Github to best estimate the efforts for a software team that is hoping to learn from the process and estimate the learning time. 

Results The validation projects are well chosen to establish that the system works with CLARA based models for brain tumor segmentation and liver transplant segmentation and with custom models such as the hip-fracture detection. 

Discussion section - Raises some valid points especially about open source software. Once again , it would be helpful to review the system reliability a year away from deployment.

Reviewer #2: Many thanks for a valuable paper on a very important topic. Templates such as described here, for effective incorporation of ML validation into clinical workflows, are essential to help overcome the serious problem of "ML in the sandbox" which never translates to the clinic. 

I see no major gaps or issues that need further substantial work. Addressing these would, I believe, increase the impact of the paper with an ML readership. 

Limitations: I am not able to assess the IT-centric aspects of the paper.

Some line-item comments follow, some trivial, some more substantial. 

The main point I would like to see better clarified is the choice landscape of cloud vs on-prem: costs, IRB constraints, time requirements. At each stage and for each element of the system, please state whether it is in-cloud or on-prem, whether this by choice or effectively required, and the pros and cons. We have found this to be a major issue to consider. If you can, please address the question of the varying levels of possible control of cloud resources, and how these can meet IRB requirements.

56: typo with period and citation (also in other spots). Also, the citations seem to lack a preceding space.

62: stray "and" before "effectiveness"

86: What is Nvidia Clara Deploy - brief description please. In general, brief orienting descriptions inline help the reader immensely.

86: What does / is XNAT? as above.

100: What does OHIF mean?

Table 1: Combining cols 1 and 2 would simplify the table structure

133: Regularize spelling of Nvidia across paper

133: Please give brief description of Triton. In particular, is it run in the cloud? This is an example of the main point described above.

144: what is REST?

Somewhere in this section: Please discuss where data is stored. Especially, are there separate channels for identified and de-identified patient data? How does data storage interact with IRB requirements, and with upload/storage costs?

Table 3, and line 187: Clara Deploy DICOM segmentation writer: Does this require collaboration with Nvidia? If so, was this by choice, and are there other options? 

183: Python (capitalized)

219-220: is clinician feedback text only, or is for example segmentation correction on images possible? please discuss the costomizability of this piece.

243 ff: Is on-site storage effectively required (for the reasons stated later)? If yes, please state this explicitly.

298: "in separate papers" is clearer (ie not separate subsections)

316: what/who is the subject of the verb "supported"

367: normalize spellings of ci^2 (at least 3 versions in paper)

382 ff: what are the general constraints on ability to import ML frameworks, eg Python? PyTorch? Matlab? Matlab bundled as an executable only? How much effort is required, eg is it just a case of configuring a docker, or is it more work. This is an important question for ML readers.

402 - 403: I believe strong and opposite arguments can be made (eg by our IT department) for choosing software with enterprise support - can you please discuss the pros and cons of open-source more deeply?

411 -452: Awesome content. Perhaps it be placed earlier in the paper, eg introduction/background. These issues are fundamental and presumably drove many of your architecture choices. In any case, readers would benefit by having these issues well in mind as they read about your proposed solution.

464: "will ultimately support any...": this is a bold claim. Is it accurate as stated?

476 - 477: See note to 402-403.

Multiple figures: in my printout they are landscape, requiring turning the paper 90 degrees. How will they be presented in the final version, and will they be legible if shrunken? Perhaps they could be better restructured to be vertical, to better use a page and allow larger fonts.

Figure 1: (a) describe Hopper (b) are inference results stored in the inference server, or is this just an interface area? See comments about storage, privacy, etc above. Are they stored as dicoms, or different file types? Are they lightweight (low space demand)?

Figure 4: bad quality/unreadable text - is a higher resolution available? 

Figure 5: Ditto - is a higher resolution version available?

Thanks much!

6. PLOS authors have the option to publish the peer review history of their article (what does this mean?). If published, this will include your full peer review and any attached files.

**Do you want your identity to be public for this peer review?** For information about this choice, including consent withdrawal, please see our Privacy Policy.

Reviewer #1: No

Reviewer #2: No

---

## [Decision Letter · Decision Letter 1]

12 Jul 2023

Implementation and prospective real-time evaluation of a generalized system for in-clinic deployment and validation of machine learning models in radiology

PDIG-D-23-00075R1

Dear Mr. Hawkins,

We are pleased to inform you that your manuscript 'Implementation and prospective real-time evaluation of a generalized system for in-clinic deployment and validation of machine learning models in radiology' has been provisionally accepted for publication in PLOS Digital Health.

Best regards,

Yuan Lai, Ph.D.

Academic Editor

PLOS Digital Health

Reviewer Comments (if any, and for reference):

Reviewer's Responses to Questions

**Comments to the Author**

1. If the authors have adequately addressed your comments raised in a previous round of review and you feel that this manuscript is now acceptable for publication, you may indicate that here to bypass the “Comments to the Author” section, enter your conflict of interest statement in the “Confidential to Editor” section, and submit your "Accept" recommendation.

Reviewer #2: All comments have been addressed

2. Does this manuscript meet PLOS Digital Health’s publication criteria? Is the manuscript technically sound, and do the data support the conclusions? The manuscript must describe methodologically and ethically rigorous research with conclusions that are appropriately drawn based on the data presented.

Reviewer #2: Yes

3. Has the statistical analysis been performed appropriately and rigorously?

Reviewer #2: N/A

4. Have the authors made all data underlying the findings in their manuscript fully available (please refer to the Data Availability Statement at the start of the manuscript PDF file)?

Reviewer #2: Yes

5. Is the manuscript presented in an intelligible fashion and written in standard English?

Reviewer #2: Yes

6. Review Comments to the Author

Reviewer #2: I believe all comments from both reviewers were clearly addressed.

The figures are much better, thanks :)

7. PLOS authors have the option to publish the peer review history of their article (what does this mean?). If published, this will include your full peer review and any attached files.

**Do you want your identity to be public for this peer review?** For information about this choice, including consent withdrawal, please see our Privacy Policy.

Reviewer #2: No
